# Preconception care utilization: Self-report versus claims-based measures among women with Medicaid

**Debra B. Stulberg**[1]*, **L. Philip Schumm**[2], **Kellie Schueler**[3], **Mihai Giurcanu**[4], **Monica E. Peek**[5]

1 Department of Family Medicine, University of Chicago, Chicago, Illinois, United States of America,
2 Center for Translational Data Science, University of Chicago, Chicago, Illinois, United States of America,
3 Department of Obstetrics, Gynecology & Reproductive Sciences, University of California San Diego, San Diego, California, United States of America, 4 Department of Public Health Sciences, University of Chicago, Chicago, Illinois, United States of America, 5 Section of General Internal Medicine, University of Chicago, Chicago, Illinois, United States of America

* Stulberg@uchicago.edu

**Data Availability Statement:** This work analyzed Pregnancy Risk Assessment and Monitoring Survey (PRAMS) data received from the U.S. Centers for Disease Control and Prevention (CDC),

## Abstract

The objective of this study is to compare self-reported preconception care utilization (PCU) among Medicaid-covered births to Medicaid claims. We identified all Medicaid-covered births to women ages 15–45 in 26 states in the year 2012 among the Pregnancy Risk Assessment and Monitoring System (PRAMS) survey and Medicaid Analytic eXtract (MAX) claims data, and identified preconception services in the latter using diagnosis codes published by Health and Human Services' Office of Population Affairs. We fit mixed-effects logistic regression models for the probability of PCU on sociodemographic factors (age, race, and ethnicity) and clinical diagnoses (depression, diabetes, or hypertension), separately for each dataset. Among 652,929 women delivering in MAX, 28.1% received at least one claims-based preconception service while an estimated 23.6% (95% CI 22.1–25.3) of PRAMS respondents reported receiving preconception care. Adjusting for age, chronic diseases, and state, PCU rates in both MAX and PRAMS were higher for non-Hispanic Black versus non-Hispanic White women (OR 1.51, 95% CI 1.49–1.54 and OR 2.05, 95% CI 1.60–2.62, respectively). Adjusting for differences in age, race and ethnicity, and state, PCU rates were higher for patients with diabetes (OR 1.34, 95% CI 1.29–1.40 and OR 1.82, 95% CI 1.16–2.85) or hypertension (OR 1.22, 95% CI 1.18–1.27 and OR 1.85, 95% CI 1.41–2.44). While Hispanic and Asian women were also more likely to report PCU than their non-Hispanic White counterparts (OR 2.07, 95% CI 1.53–2.80 and OR 3.37, 95% CI 2.28–4.98), they were less likely to have received it (OR 0.74, 95% CI 0.73–0.75 and OR 0.65, 95% CI 0.63–0.67). In conclusion, comparing self-report to claims measures of PCU, we found similar trends in the differences between non-Hispanic Black and White women, and between those with vs. without diabetes and hypertension. However, the two data sources differed in trends in other racial/ethnic groups (differences between Hispanic vs. non-Hispanic White women, and between Asian vs. non-Hispanic White women), and in those with vs. without depression. This suggests that while Medicaid claims can be a useful tool for studying preconception care, they may miss certain types of care among some sub-groups of the

and Medicaid Analytic Extract (MAX) files from the U.S. Centers for Medicare and Medicaid Services (CMS) under approved Data Use Agreements. These agreements prohibit data sharing to protect subject privacy. Researchers wishing to receive MAX data from CMS should contact the Research Data Assistance Center at www.resdac.org, and those wishing to use PRAMS data should contact the CDC at https://www.cdc.gov/prams/prams-data/researchers.htm.

**Funding:** This work was support by the Agency for Healthcare Research and Quality (R03 HS027027 to DS). The content is solely the responsibility of the authors and does not necessarily represent the official views of the National Institutes of Health. The funders had no role in study design, data collection and analysis, decision to publish, or preparation of the manuscript.

**Competing interests:** The authors have declared that no competing interests exist.

population or be subject to reporting differences that are hard to surmise. Both data sets have potential benefits and drawbacks as research tools.

## Introduction

Preconception care, defined as preventive healthcare a patient receives before pregnancy to address pregnancy-related risk factors, has been hailed as a promising strategy to improve maternal and infant outcomes [1–3]. In the United States, perinatal outcomes are markedly worse for those from minoritzed racial and ethnic backgrounds, and preconception care is one proposed strategy to reduce these disparities [4]. The rationale for preconception care is that many underlying causes of adverse pregnancy outcomes, such as maternal chronic diseases, are best addressed using a preventive approach before pregnancy [5]. While high quality pre-natal and intrapartum care are important, they may occur too late to mitigate many risk factors and to prevent maternal and infant morbidity and mortality. The Integrated Perinatal Health Framework emphasizes the role of multiple determinants on pregnancy outcomes [6]. The United States Centers for Disease Control and Prevention (CDC) currently recommends that all women of childbearing age receive preconception care [3].

There is a growing evidence base supporting the potential value of preconception care for improving health outcomes. For example, randomized controlled trials have demonstrated that preconception glycemic control among women with diabetes improves infant outcomes [7]. Preconception counseling can improve pre-pregnancy maternal behaviors such as reducing alcohol consumption and increasing folic acid intake, which are known to improve infant outcomes [7–10]. Furthermore, preconception care may reduce racial and ethnic disparities in adverse pregnancy outcomes since multiple preconception risk factors are disproportionately prevalent among women of color [11]. Accurate measures of preconception care utilization (PCU) are thus needed to assess their association with perinatal outcomes.

There is no gold standard for measuring PCU at the population level. Self-reported receipt of preconception care—both general counseling and specific preconception health services—has been assessed on several U.S. population surveys, including the CDC's Pregnancy Risk Assessment and Monitoring Systems (PRAMS) and the Behavioral Risk Factor Surveillance System (BRFSS) [12, 13]. However, for population surveillance, measures of PCU drawn from administrative sources would be beneficial. Unlike self-report that is subject to both recall bias and differences between subgroups in the way that questions are understood, administrative data have the potential advantages of using common coding systems across multiple data sources (e.g., public and private insurance). Moreover, administrative data encompass entire populations rather than relying only on samples, thereby permitting the study of rare outcomes such as severe maternal morbidity (SMM). Medicaid is publicly-funded health insurance and it covers approximately half of all U.S. births. Unlike all private insurers, Medicaid makes claims data available for research purposes. For this reason, we sought to establish the feasibility of using Medicaid claims to measure PCU and to compare this to self-reported PCU from PRAMS.

## Materials and methods

### Ethics statement

The University of Chicago Institutional Review Board (IRB) approved this study (protocol #IRB19-1291). The IRB waived the requirement for informed consent, including for subjects

<18 years of age, because this was a secondary analysis of retrospective data in which the investigators would not be identifying individual participants.

## Data

We conducted a retrospective secondary data analysis of Medicaid Analytic Extract (MAX) data files from the U.S. Centers for Medicare and Medicaid Services (CMS) from 2010–2014 under a re-use Data Use Agreement (DUA) approved on December 12, 2019. These data files were accessed immediately following DUA approval. They include person-level information on Medicaid enrollees and encounter-level information for Medicaid claims from all sources of care, including inpatient, outpatient, physician services, radiology, clinic visits, and pharmacies. We also conducted a secondary data analysis of PRAMS data from the U.S. Center for Disease Control and Prevention (CDC) from 2010–2014 under a DUA approved on November 13, 2019. Data were received and accessed on December 23, 2019. The authors were unable to identify individual participants in either data set.

From MAX files, we reviewed claims from all female beneficiaries, aged 15–45, enrolled in Medicaid from all available states (45) and Washington DC who gave birth in 2012 and had continuous Medicaid enrollment from January 2010 through December 2012. Deliveries were identified using the following International Classification of Diseases-9th revision (ICD9) diagnosis codes: V27.xx with or without 650 for normal deliveries; and V27.xx with 644.2, 644.4, 765.0 or 765.1 for preterm births. For women with more than one delivery in calendar year 2012, only information from the first delivery was used.

For each index 2012 birth, the corresponding date of conception was estimated using a modified version of the approach described by Palmsten et al. to estimate last menstrual period [14]. The date of conception was calculated to be 255 days before a full-term birth and 230 days before a premature birth. We identified preconception care in the MAX Other Therapies (OT) files using a list of 55 ICD9 codes published by the United States Department of Health and Human Services' Office of Population Affairs under its Quality Family Planning program [15]. These are classified in 7 domains of services (Table 1): contraceptive services, pregnancy testing and counseling, achieving pregnancy, basic infertility services, preconception health services, sexual transmitted diseases services, and related preventive health services. Examples of these service categories include counseling on oral contraceptive prescription (v25.01) under contraceptive services, screening for diabetes (v77.1), hypertension (v81.1), or alcoholism (v79.1) under preconception health services, and general medical exam (v70.0) under related preventive health services. If a woman had an encounter in the year prior to conception

**Table 1. ICD9 codes included in each preconception care domain.**

| Preconception care domain | ICD9 Codes |
| --- | --- |
| Contraceptive services | V25.01, V25.02, V25.03, V25.04, V25.09, V25.11, V25.12, V25.13, V25.2, V25.40, V25.41, V25.42, V25.43, V25.49, V25.5, V25.8, V25.9 |
| Pregnancy testing and counseling | V72.40, V72.41, V72.42 |
| Achieving pregnancy | V26.41 |
| Basic infertility services | 606.9, 628.0, 628.1, 628.2, 628.3, 628.4, 628.9, V26.21 |
| Preconception health services | V15.82, V26.49, V65.3, V65.41, V65.42, V77.1, V77.8, V79.0, V79.1, V81.1 |
| STD services | V01.6, V02.8, V12.09, V65.40, V65.44, V65.45, V65.5, V69.2, V73.81, V73.88, V73.89, V74.5 |
| Related preventive health services | V70.0, V72.31, V76.19, V76.2 |

that included any diagnosis code from these 7 domains, we classified her as having received preconception care. We also computed separate indicators of having received care within each of the 7 preconception care domains.

We compared the claims-based measure from MAX to patient reported receipt of preconception counseling from PRAMS among Medicaid-covered patients who experienced a delivery in 2012. We used the Phase 7 Core PRAMS Questionnaire to identify women who received Medicaid prior to pregnancy (Question 8: "During the month before you got pregnant with your new baby, what kind of health insurance did you have?"); of these, we identified those women who reported receiving preconception health counseling (Question 10: "Before you got pregnant with your new baby, did a doctor, nurse, or other health care worker talk to you about how to improve your health before pregnancy?"). Data from 26 states that administered the question about preconception counseling in 2012 were used in the data analysis. Survey weights adjusting for differences in the probability of selection and differential non-response were provided with the data, as well as information about the sampling strata; these were used in the data analysis to obtain unbiased estimates and design-based standard errors (SE).

Covariates hypothesized to be associated with the likelihood of receiving preconception care and available in both datasets included the following: (1) age (grouped into 15–17, 18–19, 20–24, 25–29, 30–34, 35–39, 40–45); (2) race and ethnicity (non-Hispanic White, non-Hispanic Black, Hispanic, Asian/Pacific Islander, and Other); and (3) presence of chronic conditions including diabetes, hypertension, and depression. Race and ethnicity were reported in both datasets, and we collapsed them into a single race/ethnicity variable for both. Chronic conditions were obtained from the chronic conditions file in MAX and self-reported by respondents to PRAMS.

## Statistical analysis

We calculated the weighted proportion of respondents reporting preconception care from PRAMS and compared this to the population proportion from MAX among the same 26 states (Fig 1A). Design-based variance estimates were computed for PRAMS using the linearization method [16] and used to construct approximate 95% confidence intervals (CIs). We also computed the population proportions receiving care within each of the 7 care domains from MAX. Finally, we compared the weighted distributions for each of the categorical covariates from PRAMS to the corresponding values from MAX.

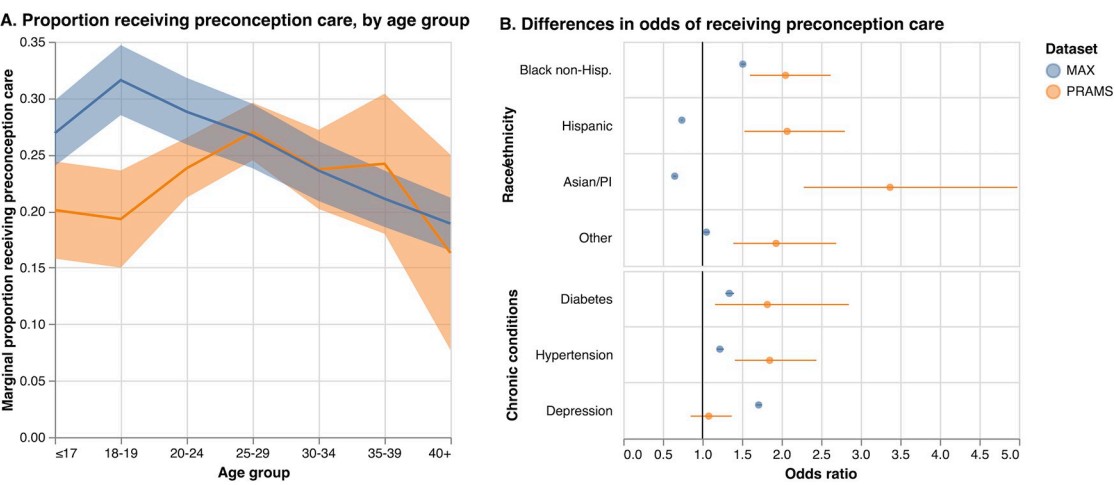

**Fig 1. Results from mixed-effects models, MAX and PRAMS 2012.**

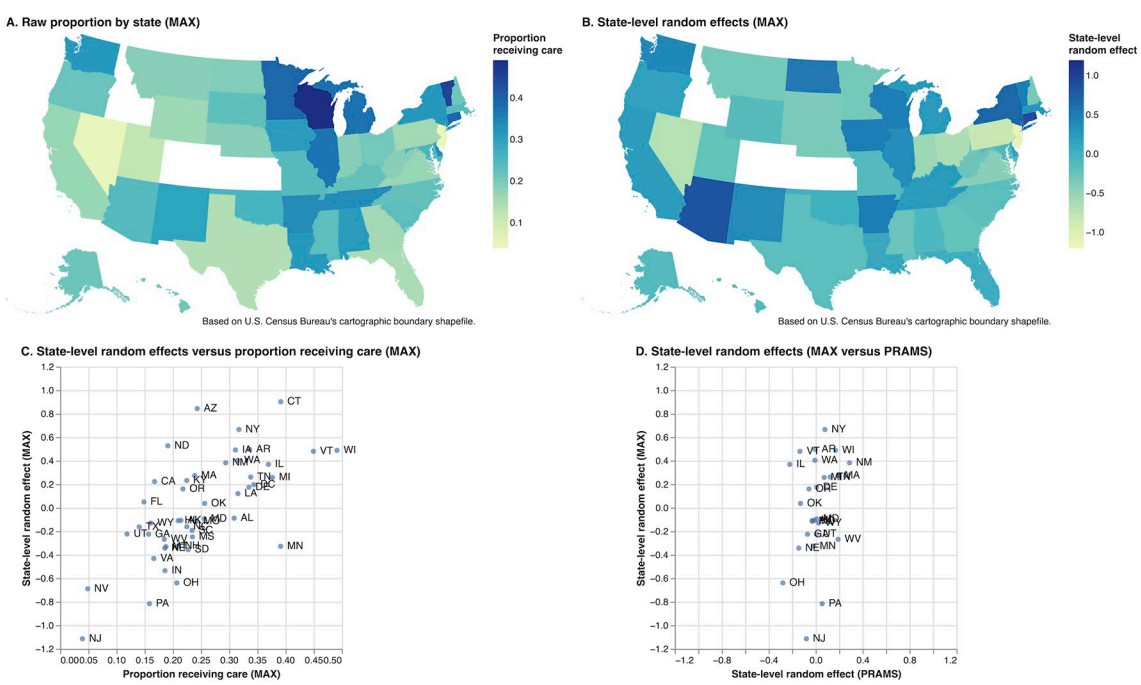

**Fig 2. Geographic variation in prevalence of preconception care utilization, MAX and PRAMS 2012.**

We fit mixed-effects logistic regression models [17] to the proportion of PCU vs. fixed effects of demographic variables (age group, and a combined race and ethnicity variable) and clinical variables (diagnosis of hypertension, diabetes, or depression) as covariates, separately for each dataset. As before, analyses of MAX were restricted to the 26 states available in PRAMS. Random effects (intercepts) were included at the state level to accommodate additional variability between states that remained after adjusting for the covariates. Since the standard mixed model assumes that the random, state-level effects are uncorrelated with the covariates, we included the state-level means of each covariate to avoid the problems that can arise if this assumption is violated [18]. Models were fit using maximum likelihood with mean–variance adaptive Gauss–Hermite quadrature [19]. Estimated fixed effects coefficients are presented as odds ratios, together with approximate 95% CIs (Fig 1B). A test of the null hypothesis that the true variance of the random effect is zero was performed using the asymptotic null distribution derived by Self and Liang [20, 21]. Marginal probabilities [22] of receiving preconception care were computed and plotted by age group for each dataset to facilitate interpretation of differences by age. Additional models excluding the random effects and including the data for all 45 states and DC (MAX only) were also fit for comparison (Table 5).

To explore geographic variability in the prevalence of PCU, we computed the population proportions (MAX) receiving care by state and the empirical Bayes means of the state-level effects and plotted these on a US map (Fig 2A and 2B), as well as against each other (Fig 2C). We also computed the empirical Bayes means of the state-level effects for PRAMS and plotted the corresponding MAX values against these (Fig 2D). All analyses were performed using Stata Release 17.0 [23].

## Results

The weighted distributions for age and for racial and ethnic groups from PRAMS were similar to those observed in MAX, although PRAMS included slightly higher proportions of women

**Table 2. Survey weighted distributions of demographic and clinical characteristics in PRAMS compared to MAX, 2012 births (percentages and 95% CI).**

| Age | PRAMS (n = 6,960)[a] | MAX (n = 652,929)[b] |
|---|---|---|
| 15–17 | 5.3 (4.6, 6.1) | 3.6 |
| 18–19 | 11.8 (10.6, 13.1) | 9.1 |
| 20–24 | 32.5 (30.7, 34.4) | 35.0 |
| 25–29 | 27.2 (25.5, 28.9) | 27.9 |
| 30–34 | 15.6 (14.3, 17.0) | 16.1 |
| 35–39 | 5.9 (5.1, 6.8) | 6.6 |
| 40+ | 1.8 (1.3, 2.3) | 1.7 |
| Race and ethnicity | | |
| Non-Hispanic White | 45.6 (44.0, 47.2) | 50.4 |
| Non-Hispanic Black | 28.1 (26.6, 29.7) | 24.9 |
| Hispanic | 16.2 (15.1, 17.3) | 17.3 |
| Asian/Pacific Islander | 5.2 (4.6, 5.9) | 4.5 |
| Other | 4.9 (4.3, 5.5) | 3.0 |
| Chronic Conditions | | |
| Diabetes | 4.0 (3.3, 4.8) | 1.6 |
| Hypertension | 7.2 (6.3, 8.3) | 2.2 |
| Depression | 15.6 (14.3, 17.0) | 7.2 |
| Preconception care | 23.6 (22.1, 25.3) | 28.1[c] |
| Contraceptive services | | 13.2 |
| Related preventive health services | | 15.4 |

[a] Includes only those who were Medicaid-eligible; survey weights and strata information provided with the dataset.

[b] Includes only those states for which PRAMS data were available.

[c] Includes all individuals with at least one claim during the year before conception in the following 7 domains: contraceptive services, pregnancy testing and counseling, achieving pregnancy, basic infertility services, preconception health services, STD services, and related preventive health services.

aged 15–20 and non-Hispanic Black women, and a slightly lower proportion of non-Hispanic White women. The prevalence of the three chronic conditions (depression, diabetes, and hypertension) based on self-report in PRAMS was higher than those based on the chronic conditions file in MAX.

Using the MAX data, we identified 1,452,034 women who delivered in 2012, of whom 23.1% received at least one health care service from the 7 domains of care within the year prior to conception. The percentage receiving preconception care varied across states (Fig 2A). Including only those living in one of the 26 PRAMS states reduced the number to 652,929 women, 28.1% of whom received preconception care (Table 2). In comparison, based on 6,960 PRAMS respondents, we estimated that 23.6% (95% CI 22.1–25.3) of Medicaid-eligible women who delivered in 2012 reported receiving preconception care; this difference was concentrated among younger women as described below.

Table 3 shows results from mixed-effects logistic regression models fit separately to both datasets. For both PRAMS and MAX, the proportion receiving preconception care declined with age after ages 25–29 and the marginal proportions for both datasets were similar, whereas for younger patients, the estimated proportions were approximately 5–10 percentage points less for PRAMS (Fig 1A). For PRAMS, all racial and ethnic minority subgroups were more likely to report having received preconception care than non-Hispanic White women, with adjusted odds ratios (OR) for non-Hispanic Black and Hispanic women relative to non-Hispanic White women of 2.05 (95% CI 1.60–2.62) and 2.07 (95% CI 1.53–2.80), respectively. In contrast, while non-Hispanic Black women in MAX were more likely than non-Hispanic

**Table 3. Mixed-effects logistic regression of preconception care measures from 2012 PRAMS and MAX vs. age, race and ethnicity, and chronic conditions.** The adjusted odds ratios (OR), their 95% confidence intervals (CI), and standard deviation (SD) of the random effects are estimated using data from 26 states available in PRAMS[a].

| Covariate | PRAMS 2012[b] | Medicaid MAX 2012[c] | | |
|---|---|---|---|---|
| | (n = 6,960) | (n = 652,929) | | |
| **Age** | Preconception care | All Domains[d] | Contraceptive services | Related prev. health services |
| 15–17 | 0.79 (0.60, 1.05) | 0.90‡ (0.87, 0.93) | 1.04* (1.00, 1.08) | 0.54‡ (0.52, 0.57) |
| 18–19 | 0.75 (0.55, 1.04) | 1.15‡ (1.13, 1.18) | 1.21‡ (1.19, 1.25) | 0.88‡ (0.86, 0.90) |
| 20–24 | Ref. | Ref. | Ref. | Ref. |
| 25–29 | 1.19 (0.96, 1.48) | 0.89‡ (0.88, 0.90) | 0.80‡ (0.79, 0.82) | 1.10‡ (1.09, 1.12) |
| 30–34 | 0.99 (0.72, 1.37) | 0.74‡ (0.73, 0.76) | 0.57‡ (0.56, 0.59) | 1.02 (1.00, 1.04) |
| 35–39 | 1.02 (0.69, 1.52) | 0.64‡ (0.62, 0.66) | 0.42‡ (0.40, 0.43) | 0.93‡ (0.90, 0.96) |
| 40+ | 0.61 (0.31, 1.22) | 0.55‡ (0.53, 0.58) | 0.28‡ (0.26, 0.31) | 0.86‡ (0.81, 0.91) |
| **Race and ethnicity** | | | | |
| Non-Hispanic White | Ref. | Ref. | Ref. | Ref. |
| Non-Hispanic Black | 2.05‡ (1.60, 2.62) | 1.51‡ (1.49, 1.54) | 1.42‡ (1.39, 1.44) | 1.27‡ (1.25, 1.29) |
| Hispanic | 2.07‡ (1.53, 2.80) | 0.74‡ (0.73, 0.75) | 0.85‡ (0.83, 0.87) | 0.71‡ (0.70, 0.73) |
| Asian/Pacific Islander | 3.37‡ (2.28, 4.98) | 0.65‡ (0.63, 0.67) | 0.47‡ (0.44, 0.49) | 0.75‡ (0.72, 0.78) |
| Other | 1.93‡ (1.39, 2.69) | 1.05† (1.02, 1.09) | 1.03 (0.99, 1.08) | 0.87‡ (0.83, 0.91) |
| **Chronic Conditions** | | | | |
| Diabetes | 1.82† (1.16, 2.85) | 1.34‡ (1.29, 1.40) | 1.04 (0.98, 1.11) | 1.38‡ (1.31, 1.44) |
| Hypertension | 1.85‡ (1.41, 2.44) | 1.22‡ (1.18, 1.27) | 1.10‡ (1.04, 1.15) | 1.24‡ (1.19, 1.29) |
| Depression | 1.08 (0.85, 1.37) | 1.71‡ (1.68, 1.75) | 1.52‡ (1.49, 1.56) | 1.52‡ (1.48, 1.55) |
| **State-level random effect** | | | | |
| SD | 0.13 (0.10, 0.17) | 0.32‡ (0.25, 0.42) | 0.42‡ (0.32, 0.55) | 0.26‡ (0.20, 0.35) |

* p < 0.05

† p < 0.01

‡ p < 0.001

[a] Models include state-level means of all covariates.

[b] Includes only those who were Medicaid-eligible.

[c] Includes only those states for which PRAMS data were available.

[d] Includes all individuals with at least one claim during the year before conception in the following 7 domains: contraceptive services, pregnancy testing and counseling, natural family planning counseling (V26.41), basic infertility services, preconception health services, STD services, and related preventive health services.

White women to have received preconception care (OR 1.51, 95% CI 1.49–1.54), Hispanic women and Asian women were *less likely* to have received preconception care than their non-Hispanic White counterparts (OR 0.74, 95% CI 0.73–0.75 and OR 0.65, 95% CI 0.63–0.67, respectively). Women with diabetes and hypertension were more likely to have received preconception care in both datasets though the differences were smaller in MAX, whereas while women with depression were also more likely to have received preconception care in MAX, there was no evidence of a higher proportion in PRAMS.

Results from logistic regressions excluding the random state-level effects were similar (Table 4).

Roughly similar patterns were observed in MAX for the odds of having received contraceptive services and related preventive health services, both domains of preconception care associated with decreased risk of severe maternal morbidity [24]. Notable differences included somewhat different patterns with age, and negligible or smaller differences in the odds of having received contraceptive services for those with diabetes or hypertension vs. those without (OR 1.04, 95% CI 0.98–1.11 and OR 1.10, 95% CI 1.04–1.15, respectively).

**Table 4. Logistic regressions of preconception care measures from 2012 PRAMS and Medicaid MAX on age, race/ethnicity and chronic conditions, estimated using data from 26 states available in PRAMS (odds ratios and 95% CI).**

| Covariate | PRAMS 2012[a] | Medicaid MAX 2012[b] | | |
|---|---|---|---|---|
| | (n = 6,960) | (n = 652,929) | | |
| Age | Preconception care | All Domains[c] | Contraceptive services | Related prev. health services |
| ≤17 | 0.79 (0.53, 1.20) | 0.89‡ (0.86, 0.92) | 1.06† (1.02, 1.10) | 0.53‡ (0.50, 0.55) |
| 18–19 | 0.75 (0.54, 1.03) | 1.16‡ (1.13, 1.18) | 1.24‡ (1.22, 1.27) | 0.88‡ (0.85, 0.90) |
| 20–24 | Ref. | Ref. | Ref. | Ref. |
| 25–29 | 1.17 (0.92, 1.48) | 0.90‡ (0.89, 0.91) | 0.80‡ (0.78, 0.81) | 1.13‡ (1.11, 1.15) |
| 30–34 | 0.98 (0.75, 1.29) | 0.76‡ (0.75, 0.77) | 0.56‡ (0.55, 0.58) | 1.06‡ (1.04, 1.08) |
| 35–39 | 1.01 (0.68, 1.49) | 0.65‡ (0.64, 0.67) | 0.40‡ (0.39, 0.42) | 0.98 (0.96, 1.01) |
| 40+ | 0.62 (0.33, 1.19) | 0.57‡ (0.55, 0.60) | 0.27‡ (0.25, 0.30) | 0.93* (0.88, 0.98) |
| Race/ethnicity | | | | |
| Non-Hispanic White | Ref. | Ref. | Ref. | Ref. |
| Non-Hispanic Black | 1.99‡ (1.58, 2.50) | 1.50‡ (1.48, 1.52) | 1.30‡ (1.28, 1.32) | 1.36‡ (1.34, 1.38) |
| Hispanic | 2.01‡ (1.55, 2.61) | 0.84‡ (0.83, 0.85) | 0.87‡ (0.85, 0.89) | 0.89‡ (0.88, 0.91) |
| Asian/Pacific Islander | 3.08‡ (2.20, 4.33) | 0.79‡ (0.77, 0.82) | 0.49‡ (0.47, 0.52) | 1.00 (0.97, 1.04) |
| Other | 1.97‡ (1.41, 2.76) | 1.21‡ (1.17, 1.25) | 1.43‡ (1.37, 1.48) | 0.87‡ (0.83, 0.91) |
| Chronic Conditions | | | | |
| Diabetes | 1.87† (1.19, 2.94) | 1.37‡ (1.31, 1.43) | 1.06 (1.00, 1.12) | 1.41‡ (1.35, 1.48) |
| Hypertension | 1.87† (1.31, 2.67) | 1.25‡ (1.21, 1.30) | 1.10‡ (1.05, 1.16) | 1.28‡ (1.23, 1.33) |
| Depression | 1.08 (0.83, 1.40) | 1.82‡ (1.79, 1.86) | 1.61‡ (1.57, 1.65) | 1.64‡ (1.60, 1.68) |

* p < 0.05

† p < 0.01

‡ p < 0.001

[a] Includes only those who were Medicaid-eligible.

[b] Includes only those states for which PRAMS data were available.

[c] Includes all individuals with at least one claim during the year before conception in the following 7 domains: contraceptive services, pregnancy testing and counseling, natural family planning counseling (V26.41), basic infertility services, preconception health services, STD services, and related preventative health services.

In MAX, the standard deviation (SD) of the state-level random effects in the model for all preconception care domains was 0.32 (95% CI 0.25–0.42), corresponding to only 3% of the total variance after adjusting for the covariates. The SD was slightly higher when fit to the data for all 45 MAX states plus DC at 0.42 (95% CI 0.34–0.52) (Table 5). Estimates of the state-specific random effects are plotted in Fig 2B; the correlation between these estimates and the state-specific prevalence of preconception care was 0.67 (Fig 2C). The SD of the random effects for PRAMS was smaller at 0.13 (95% CI 0.10–0.17), and a test of the null hypothesis that the state level variance component is zero was not significant (p = 0.27). Among the 26 PRAMS states, the correlation between the PRAMS state-specific effects and those from MAX was only 0.31 (Fig 2D).

## Discussion

In this retrospective secondary data analysis, we compared a Medicaid claims-based measure of PCU to women's self-report from the PRAMS survey. Overall, we found similar rates of PCU in both datasets. This has important implications for health services research, which often weighs the benefits and limitations of using claims and survey data in study designs. For measuring PCU, Medicaid claims data have several important features that overcome limitations of the PRAMS data. These limitations include lack of specificity about services received,

**Table 5. Mixed-effects logistic regressions of preconception care measures from 2012 Medicaid MAX on age, race/ethnicity and chronic conditions, estimated using data from 46 states (odds ratios and 95% CI)[a].**

| Covariate | Medicaid MAX 2012[b] (n = 1,452,034) | | |
|---|---|---|---|
| Age | All Domains[c] | Contraceptive services | Related prev. health services |
| ≤17 | 0.89‡ (0.87, 0.91) | 0.91‡ (0.88, 0.93) | 0.61‡ (0.59, 0.63) |
| 18–19 | 1.19‡ (1.17, 1.20) | 1.16‡ (1.14, 1.18) | 1.02 (1.00, 1.04) |
| 20–24 | Ref. | Ref. | Ref. |
| 25–29 | 0.85‡ (0.84, 0.85) | 0.79‡ (0.78, 0.80) | 1.05‡ (1.04, 1.06) |
| 30–34 | 0.69‡ (0.68, 0.70) | 0.57‡ (0.56, 0.57) | 0.95‡ (0.93, 0.96) |
| 35–39 | 0.58‡ (0.57, 0.59) | 0.41‡ (0.40, 0.42) | 0.87‡ (0.85, 0.89) |
| 40+ | 0.50‡ (0.49, 0.52) | 0.29‡ (0.27, 0.31) | 0.83‡ (0.79, 0.86) |
| Race/ethnicity | | | |
| Non-Hispanic White | Ref. | Ref. | Ref. |
| Non-Hispanic Black | 1.60‡ (1.58, 1.61) | 1.50‡ (1.48, 1.52) | 1.38‡ (1.36, 1.40) |
| Hispanic | 0.70‡ (0.69, 0.70) | 0.77‡ (0.76, 0.78) | 0.69‡ (0.68, 0.70) |
| Asian/Pacific Islander | 0.67‡ (0.65, 0.68) | 0.53‡ (0.51, 0.55) | 0.79‡ (0.77, 0.81) |
| Other | 1.15‡ (1.12, 1.18) | 1.13‡ (1.10, 1.17) | 0.86‡ (0.83, 0.89) |
| Chronic Conditions | | | |
| Diabetes | 1.41‡ (1.36, 1.45) | 1.12‡ (1.07, 1.17) | 1.47‡ (1.41, 1.52) |
| Hypertension | 1.27‡ (1.24, 1.31) | 1.14‡ (1.10, 1.19) | 1.31‡ (1.27, 1.36) |
| Depression | 1.74‡ (1.71, 1.77) | 1.56‡ (1.53, 1.59) | 1.60‡ (1.57, 1.63) |
| State-level random effect | | | |
| SD | 0.42‡ (0.34, 0.52) | 0.48‡ (0.39, 0.59) | 0.47‡ (0.38, 0.57) |

* p < 0.05

† p < 0.01

‡ p < 0.001

[a] Models include state-level means of all covariates.

[b] Includes only those states for which PRAMS data were available.

[c] Includes all individuals with at least one claim during the year before conception in the following 7 domains: contraceptive services, pregnancy testing and counseling, natural family planning counseling (V26.41), basic infertility services, preconception health services, STD services, and related preventative health services.

lack of information about reproductive history, and narrowly including only women with a recent live birth [25–27]. In addition, PRAMS does not assess rare but serious maternal complications such as severe maternal morbidity [28], which would be difficult to identify through self-report but can be quantified and assessed for association with preconception services using Medicaid claims as well as present in sufficient numbers to permit quantitative analyses [24]. Further, PRAMS asks postpartum women to recall if a healthcare provider counseled them about their health prior to pregnancy, which may be subject to recall bias. Because of these inherent limitations in using surveys for tracking preconception care, it is encouraging to see claims-based measures offer a promising alternative for population-wide surveillance. Nonetheless, claims data also have important limitations, such as the fact that while diagnosis and procedures codes capture an array of billed services, they do not capture the content or quality of counseling during encounters. In addition, variation in provider billing practices and state reporting may generate additional variation that may have little or no relation to subsequent outcomes. This may account for the additional state-level variability observed in MAX but not in PRAMS.

We found replicable associations between PCU and two key constructs: race and ethnicity (PCU was higher among non-Hispanic Black women than among non-Hispanic White

women) and the presence of chronic conditions (PCU was more common among patients with diabetes or hypertension, as expected). While we were able to adjust for diabetes, hypertension and depression, a limitation of the study is that we were not able to adjust for other chronic conditions for which women might seek routine medical care before pregnancy, such as obesity or frequent mental distress, that are more prevalent among non-Hispanic Black than non-Hispanic White women [11]. These may account, at least in part, for some of the remaining difference in PCU between non-Hispanic Black and White women. Other reasons for the high rate among non-Hispanic Black women may be a high underlying desire to seek preventive care when lack of insurance and cost are not barriers [29]. Interestingly, Hispanic and Asian women were also substantially more likely to report PCU despite having *lower* odds of receiving such services based on MAX data. This paradox may reflect socio-cultural differences in the way that the single item question in PRAMS is understood and/or in the interactions between health care providers and patients. Prior research has also found that Hispanic women were less likely than non-Hispanic White women to have stable Medicaid coverage (OR 0.6, 95% CI 0.5–0.7) compared to White women [30], which could cause under-counting of preconception care services received. These are additional limitations of our study. Finally, the quality of race and ethnicity data in Medicaid has been found to vary across states [31].

We also found that women under 25 years of age were more likely to underreport PCU relative to the MAX claims than women over 25 years old. This may reflect the wording of the item in PRAMS which asked patients if their clinicians talked about how to improve their health *before pregnancy*. Very young women may be less likely to view preventive healthcare as "pre-pregnancy" care. They may also be less likely to receive such services outside of routine gynecological care because of the lower prevalence of chronic diseases, and thus be less likely to report having received preconception services per se. This is consistent with the data from MAX showing that the rate of related preventive health services is lower for women under 20 while that for contraceptive services is higher.

## Conclusions

PRAMS provides an important basis for obtaining population prevalence estimates of PCU among women who have given birth [25–27]. Researchers have used PRAMS to compare PCU between racial and ethnic groups and to track improvements in women's health and healthcare following the Affordable Care Act [26]. As we have shown here, Medicaid claims-based measures provide new opportunities to study PCU in this population while addressing many of the limitations of self-report data based on samples of the population.

Our results suggest that it is critical for researchers using either source of data to study PCU to carefully consider both the potential benefits and limitations of the source in order to interpret their results correctly. Claims-based estimates of PCU demonstrate expected associations with certain demographic (e.g., age and differences between non-Hispanic Black and White women) and clinical characteristics while being less subject to recall bias, suggesting that they can be used to study certain factors associated with PCU and its effects on outcomes. At the same time, it is possible that claims-based estimates may miss certain types of care among some subgroups, as well as be affected by reporting differences between states that would limit their utility for between-state comparisons. Researchers using these two approaches to quantifying preconception care should consider the potential benefits and limitations of each.

## Acknowledgments

The authors gratefully acknowledge Ashley McHugh for contributions to data acquisition and project management and Angel Boulware for literature search.

## Author Contributions

**Conceptualization:** Debra B. Stulberg.

**Data curation:** Kellie Schueler, Mihai Giurcanu.

**Formal analysis:** L. Philip Schumm, Kellie Schueler, Mihai Giurcanu.

**Funding acquisition:** Debra B. Stulberg, L. Philip Schumm.

**Methodology:** L. Philip Schumm, Mihai Giurcanu, Monica E. Peek.

**Writing – original draft:** Debra B. Stulberg, L. Philip Schumm.

**Writing – review & editing:** Kellie Schueler, Mihai Giurcanu, Monica E. Peek.

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
