## [Decision Letter · Decision Letter 0]

5 Oct 2023

PGPH-D-23-01734

Preconception Care Utilization: Self-report versus Claims-based Measures among Women with Medicaid

Dear Dr. Stulberg,

Thank you for submitting your manuscript to PLOS Global Public Health. After careful consideration, we feel that it has merit but does not fully meet PLOS Global Public Health’s publication criteria as it currently stands. Therefore, we invite you to submit a revised version of the manuscript that addresses the points raised during the review process.

We look forward to receiving your revised manuscript.

Kind regards,

Nebiyu Dereje, MPH, PhD

Academic Editor

Journal Requirements:

2. Please provide separate figure files in .tif or .eps format only and remove any figures embedded in your manuscript file. Please also ensure all files are under our size limit of 10MB.

3. Some material included in your submission may be copyrighted. According to PLOS’s copyright policy, authors who use figures or other material (e.g., graphics, clipart, maps) from another author or copyright holder must demonstrate or obtain permission to publish this material under the Creative Commons Attribution 4.0 International (CC BY 4.0) License used by PLOS journals. Please closely review the details of PLOS’s copyright requirements here: PLOS Licenses and Copyright. If you need to request permissions from a copyright holder, you may use PLOS's Copyright Content Permission form.

Potential Copyright Issues:

Fig 1: please (a) provide a direct link to the base layer of the map (i.e., the country or region border shape) and ensure this is also included in the figure legend; and (b) provide a link to the terms of use / license information for the base layer image or shapefile. We cannot publish proprietary or copyrighted maps (e.g. Google Maps, Mapquest) and the terms of use for your map base layer must be compatible with our CC-BY 4.0 license. 

"

Additional Editor Comments (if provided):

1. Please include 95% CI for the proportion estimates in the abstract --to indicate the difference in the two groups.

2. Was the association between variables adjusted for confounders? How was handled?

3. The reported OR is it adjusted or not?

4. What are the limitations of this study?

Reviewer 1:

This is a great paper covering a very important topic, especially on some marginalized populations in the States. The authors have done a terrific job putting the paper together in a good academic fashion with good written English. However, there are some comments I would like to make.

1. Abstract: under the conclusion subheading. This summary is unclear. Can you add a bit more detail, just a short one

2. Line 208, fix the verb “was”

3. Discussion: Can you explain why PCU rates in MAX and PRAMS were higher for non-Hispanic Black versus non-Hispanic White women? Give reference for your claims

4. Line 35, Discussion: In the previous section, for example, in Table 2, you mentioned Asian and Pacific Women of origin. Is this a different cohort?

5. Line 51-54, Discussion: Can you summarize your claims with statistics and references?

Reviewer 2:

This is a very important topic for discussion and the authors did a good job with this manuscript. The various components are well written and the available data elaborates what is expected to be seen at the population level.

Reviewers' comments:

Reviewer's Responses to Questions

**Comments to the Author**

1. Does this manuscript meet PLOS Global Public Health’s publication criteria? Is the manuscript technically sound, and do the data support the conclusions? The manuscript must describe methodologically and ethically rigorous research with conclusions that are appropriately drawn based on the data presented.

Reviewer #1: Yes

Reviewer #2: Yes

2. Has the statistical analysis been performed appropriately and rigorously?

Reviewer #1: Yes

Reviewer #2: Yes

3. Have the authors made all data underlying the findings in their manuscript fully available (please refer to the Data Availability Statement at the start of the manuscript PDF file)?

Reviewer #1: Yes

Reviewer #2: Yes

4. Is the manuscript presented in an intelligible fashion and written in standard English?

Reviewer #1: Yes

Reviewer #2: Yes

5. Review Comments to the Author

Reviewer #1: This is a great paper covering a very important topic, especially on some marginalized populations in the States. The authors have done a terrific job putting the paper together in a good academic fashion with good written English. However, there are some comments I would like to make.

1. Abstract: under the conclusion subheading. This summary is unclear. Can you add a bit more detail, just a short one

2. Line 208, fix the verb “was”

3. Discussion: Can you explain why PCU rates in MAX and PRAMS were higher for non-Hispanic Black versus non-Hispanic White women? Give reference for your claims

4. Line 35, Discussion: In the previous section, for example, in Table 2, you mentioned Asian and Pacific Women of origin. Is this a different cohort?

5. Line 51-54, Discussion: Can you summarize your claims with statistics and references?

Reviewer #2: This is a very important topic for discussion and the authors did a good job with this manuscript. The various components are well written and the available data elaborates what is expected to be seen at the population level.

6. PLOS authors have the option to publish the peer review history of their article (what does this mean?). If published, this will include your full peer review and any attached files.

**Do you want your identity to be public for this peer review?** For information about this choice, including consent withdrawal, please see our Privacy Policy.

Reviewer #1: **Yes: **Lydia Kaforau

Reviewer #2: No

---

## [Editor Report · Decision Letter 1]

8 Nov 2023

Preconception Care Utilization: Self-report versus Claims-based Measures among Women with Medicaid

PGPH-D-23-01734R1

Dear Dr. Stulberg,

We are pleased to inform you that your manuscript 'Preconception Care Utilization: Self-report versus Claims-based Measures among Women with Medicaid' has been provisionally accepted for publication in PLOS Global Public Health.

Best regards,

Nebiyu Dereje, MPH, PhD

Academic Editor